# ON THE OPTIMAL MEMORIZATION POWER OF ReLU NEURAL NETWORKS

**Gal Vardi,**[*] **Gilad Yehudai**[*] **& Ohad Shamir**
Department of Computer Science
Weizmann Institute of Science
{gal.vardi,gilad.yehudai,ohad.shamir}@weizmann.ac.il

## ABSTRACT

We study the memorization power of feedforward ReLU neural networks. We show that such networks can memorize any $N$ points that satisfy a mild separability assumption using $\tilde{O}\left(\sqrt{N}\right)$ parameters. Known VC-dimension upper bounds imply that memorizing $N$ samples requires $\Omega(\sqrt{N})$ parameters, and hence our construction is optimal up to logarithmic factors. We also give a generalized construction for networks with depth bounded by $1 \leq L \leq \sqrt{N}$, for memorizing $N$ samples using $\tilde{O}(N/L)$ parameters. This bound is also optimal up to logarithmic factors. Our construction uses weights with large bit complexity. We prove that having such a large bit complexity is both necessary and sufficient for memorization with a sub-linear number of parameters.

## 1 INTRODUCTION

The expressive power of neural networks has been widely studied in many previous works. These works study different aspects of expressiveness, such as the universal approximation property (Cybenko, 1989; Leshno et al., 1993), and the benefits of depth in neural networks (Telgarsky, 2016; Eldan and Shamir, 2016; Safran and Shamir, 2017; Daniely, 2017; Chatziafratis et al., 2019). Another central and well studied question is about their memorization power.

The problem of memorization in neural networks can be viewed in the following way: For every dataset of $N$ labeled samples $(\mathbf{x}_1, y_1), \ldots, (\mathbf{x}_N, y_N) \in \mathcal{X} \times \mathcal{Y}$, construct a network $\mathcal{N} : \mathcal{X} \to \mathcal{Y}$ such that $\mathcal{N}(\mathbf{x}_i) = y_i$ for every $i = 1, \ldots, N$. Many works have shown results regarding the memorization power of neural networks, using different assumptions on the activation function and data samples (see e.g. Huang and Babri (1998); Huang (2003); Baum (1988); Vershynin (2020); Daniely (2019; 2020); Bubeck et al. (2020); Park et al. (2020); Hardt and Ma (2016); Yun et al. (2019); Zhang et al. (2021); Nguyen and Hein (2018); Rajput et al. (2021); Sontag (1997)). The question of memorization also have practical implications on phenomenons such as "double descent" (Belkin et al. (2019); Nakkiran et al. (2019)) which connects the memorization power of neural networks with their generalization capabilities.

A trivial lower bound on the required size of the network for memorizing $N$ labeled points is implied by the VC dimension of the network (cf. Shalev-Shwartz and Ben-David (2014) ). That is, if a network with a certain size cannot shatter any *specific* set of $N$ points, then it certainly cannot memorize *all* sets of $N$ points. Known VC dimension bounds for networks with $W$ parameters is on the order of $O(W^2)$ (Goldberg and Jerrum (1995); Bartlett et al. (1998; 2019)). Hence, it follows that memorizing $N$ samples would require at least $\Omega\left(N^{1/2}\right)$ parameters. The best known upper bound is given in Park et al. (2020), where it is shown that memorizing $N$ data samples can be done using a neural network with $O\left(N^{2/3}\right)$ parameters. Thus, there is a clear gap between the lower and upper bounds, although we note that the upper bound is for memorization of *any* set of data samples, while the lower bound is for shattering a single set of data samples. In this paper we ask the following questions:

---

[*]Equal contribution

> *What is the minimal number of parameters that are required to memorize $N$ labeled data samples? Is the task of memorizing any set of $N$ data samples more difficult than shattering a single set of $N$ samples?*

We answer these questions by providing a construction of a ReLU feedforward neural network which achieves the lower bound up to logarithmic factors. In this construction we use a very deep neural network, but with a constant width of 12. In more details, our main result is the following:

**Theorem 1.1** (informal statement). *Let* $(\mathbf{x}_1, y_1), \ldots, (\mathbf{x}_N, y_N) \in \mathbb{R}^d \times \{1, \ldots, C\}$ *be a set of $N$ labeled samples of a constant dimension $d$, with $\|\mathbf{x}_i\| \leq r$ for every $i$ and $\|\mathbf{x}_i - \mathbf{x}_j\| \geq \delta$ for every $i \neq j$. Then, there exists a ReLU neural network $F : \mathbb{R}^d \to \mathbb{R}$ with width 12, depth $\tilde{O}\left(\sqrt{N}\right)$, and $\tilde{O}\left(\sqrt{N}\right)$ parameters, such that $F(\mathbf{x}_i) = y_i$ for every $i \in [N]$, where the notation $\tilde{O}(\cdot)$ hides logarithmic factors in $N, C, r, \delta^{-1}$.*

Comparing this result to the known VC bounds, we show that, up to logarithmic factors, our construction is optimal. This also shows, quite surprisingly, that up to logarithmic factors, the task of memorizing any set of $N$ points is not more difficult than shattering a single set of $N$ points, under the mild separability assumption of the data samples. We note that this result can also be extended to regression tasks (see Remark 3.3).

In our construction, the depth of the network is $\tilde{\Theta}\left(\sqrt{N}\right)$. We also give a generalized construction where the depth of the network is limited to some $1 \leq L \leq \sqrt{N}$. In this case, the number of parameters in our construction is $\tilde{O}\left(\frac{N}{L}\right)$ (see Theorem 5.1). We compare this result to the VC-dimension bound from Bartlett et al. (2019), and show that our construction is optimal up to logarithmic factors.

Our construction uses a bit extraction technique, inspired by Telgarsky's triangle function (Telgarsky (2016)), and by Safran and Shamir (2017). Using this technique, we are able to use weights with bit complexity $\tilde{\Theta}(\sqrt{N})$, and deep neural networks to "extract" the bits of information from the specially crafted weights of the network. We also generalize our results to the case of having a bounded bit complexity restriction on the weights. We show both lower (Theorem 6.1) and upper (Theorem 6.2) bounds, proving that memorizing $N$ points using a network with $N^{1-\epsilon}$ parameters, for some $\epsilon \in [0, 0.5]$ can be done if the bit complexity of each weight is $\tilde{\Theta}\left(N^{\epsilon}\right)$. Hence, our construction is also optimal, up to logarithmic factors, w.r.t the bit complexity of the network. We emphasize that also in previous works showing non-trivial VC bounds (e.g. Bartlett et al. (1998; 2019)) weights with large bit complexity are used. We note that increasing the bit complexity beyond $N^{1/2}$ cannot be used to further reduce the number of parameters (see the discussion in section 4).

### RELATED WORK

#### MEMORIZATION – UPPER BOUNDS

The problem of memorizing arbitrary data points with neural networks has a rich history. Baum (1988) studied memorization in single-hidden-layer neural networks with the threshold activation, and showed that $\lceil \frac{N}{d} \rceil$ neurons suffice to memorize $N$ arbitrary points in general position in $\mathbb{R}^d$ with binary labels. Bubeck et al. (2020) extended the construction of Baum (1988) and showed that single-hidden-layer ReLU networks with $4 \cdot \lceil \frac{N}{d} \rceil$ hidden neurons can memorize $N$ points in general position with arbitrary real labels. In Huang et al. (1991) and Sartori and Antsaklis (1991) it is shown that single-hidden-layer networks with the threshold activation can memorize any arbitrary set of $N$ points, even if they are not in general position, using $N - 1$ neurons. Huang and Babri (1998) proved a similar result for any bounded non-linear activation function $\sigma$ where either $\lim_{z \to \infty} \sigma(z)$ or $\lim_{z \to -\infty} \sigma(z)$ exists. Zhang et al. (2021) proved that single-hidden-layer ReLU networks can memorize arbitrary $N$ points in $\mathbb{R}^d$ with arbitrary real labels using $N$ neurons and $2N + d$ parameters.

Huang (2003) showed that two-hidden-layers networks with the sigmoid activation can memorize $N$ points with $O(\sqrt{N})$ neurons, but the number of parameters is still linear in $N$. Yun et al. (2019) proved a similar result for ReLU (and hard-tanh) networks. Vershynin (2020) showed that threshold and ReLU networks can memorize $N$ binary-labeled unit vectors in $\mathbb{R}^d$ separated by a distance of $\delta > 0$, using $\tilde{O}\left(e^{1/\delta^2} + \sqrt{N}\right)$ neurons and $\tilde{O}\left(e^{1/\delta^2}(d + \sqrt{N}) + N\right)$ parameters. Rajput et al.

(2021) improved the dependence on $\delta$ by giving a construction with $\tilde{O}\left(\frac{1}{\delta} + \sqrt{N}\right)$ neurons and $\tilde{O}\left(\frac{d}{\delta} + N\right)$ parameters. This result holds only for threshold networks, but does not assume that the inputs are on the unit sphere.

The memorization power of more specific architectures was also studied. Hardt and Ma (2016) proved that residual ReLU networks with $O(N)$ neurons can memorize $N$ points on the unit sphere separated by a constant distance. Nguyen and Hein (2018) considered convolutional neural networks and showed, under certain assumptions, memorization using $O(N)$ neurons.

Note that in all the results mentioned above the number of parameters is at least linear in $N$. Our work is inspired by Park et al. (2020), that established a first memorization result with a sub-linear number of parameters. They showed that neural networks with sigmoidal or ReLU activations can memorize $N$ points in $\mathbb{R}^d$ separated by a normalized distance of $\delta$, using $O\left(N^{2/3} + \log(1/\delta)\right)$ parameters (where the dimension $d$ is constant). Thus, in this work we improve the dependence on $N$ from $N^{2/3}$ to $\sqrt{N}$ (up to logarithmic factors), which is optimal. We also note that the first stage in our construction is similar to the first stage in theirs. In Park et al. (2020) there is also a discussion about the bit complexity of the network, and the necessity that the bit complexity depends on $N$ to achieve memorization with a sub-linear number of parameters.

Finally, optimization aspects of memorization were studied in Bubeck et al. (2020); Daniely (2019; 2020).

### MEMORIZATION – LOWER BOUNDS

Lower bounds on the number of parameters required for memorization are implied by bounds on the VC dimension of neural networks. Indeed, if $W$ parameters are not sufficient for shattering even a single set of size $N$, then they are clearly not sufficient for memorizing all sets of size $N$. The VC dimension of neural networks has been extensively studied in recent decades (cf. Anthony and Bartlett (2009); Bartlett et al. (2019)). The most relevant results for our work are by Goldberg and Jerrum (1995) and Bartlett et al. (2019). We discuss these results and their implications in Sections 4 and 5.

Trade-offs between the number of parameters of the network and the Lipschitz parameter of the prediction function in memorizing a given dataset are studies in Bubeck et al. (2021); Bubeck and Sellke (2021).

### THE BENEFITS OF DEPTH

In this work we show that deep networks have significantly more memorization power. Quite a few theoretical works in recent years have explored the beneficial effect of depth on increasing the expressiveness of neural networks (e.g., Martens et al. (2013); Eldan and Shamir (2016); Telgarsky (2016); Liang and Srikant (2016); Daniely (2017); Safran and Shamir (2017); Yarotsky (2017); Safran et al. (2019); Chatziafratis et al. (2019); Vardi and Shamir (2020); Bresler and Nagaraj (2020); Venturi et al. (2021); Vardi et al. (2021)). The benefits of depth in the context of the VC dimension is implied by, e.g., Bartlett et al. (2019). Finally, Park et al. (2020) already demonstrated that deep networks have more memorization power than shallow ones, albeit with a weaker bound than ours.

## 2 PRELIMINARIES

### NOTATIONS

For $n \in \mathbb{N}$ and $i \leq j$ we denote by $\text{BIN}_{i:j}(n)$ the string of bits in places $i$ until $j$ inclusive, in the binary representation of $n$ and treat it as an integer (in binary basis). For example, $\text{BIN}_{1:3}(32) = 4$. We denote by $\text{LEN}(n)$ the number of bits in its binary representation. We denote $\text{BIN}_i(n) := \text{BIN}_{i:i}(n)$. For a function $f$ and $i \in \mathbb{N}$ we denote by $f^{(i)}$ the composition of $f$ with itself $i$ times. We denote vectors in bold face. We use the $\tilde{O}(N)$ notation to hide logarithmic factors, and use $O(N)$ to hide constant factors. For $n \in \mathbb{N}$ we denote $[n] := \{1, \ldots, n\}$. We say that a hypothesis class $\mathcal{H}$ *shatters* the points $\mathbf{x}_1, \ldots, \mathbf{x}_N \in \mathcal{X}$ if for every $y_1, \ldots, y_N \in \{\pm 1\}$ there is $h \in \mathcal{H}$ s.t. $h(\mathbf{x}_i) = y_i$ for every $i = 1, \ldots, N$.

NEURAL NETWORKS

We denote by $\sigma(z) := \max\{0, z\}$ the ReLU function. In this paper we only consider neural networks with the ReLU activation.

Let $d \in \mathbb{N}$ be the data input dimension. We define a *neural network* of depth $L$ as $\mathcal{N} : \mathbb{R}^d \to \mathbb{R}$ where $\mathcal{N}(\mathbf{x})$ is computed recursively by

- $\mathbf{h}^{(1)} = \sigma\left(W^{(1)}\mathbf{x} + \mathbf{b}^{(1)}\right)$ for $W^{(1)} \in \mathbb{R}^{m_1 \times d}, \mathbf{b}^{(1)} \in \mathbb{R}^{m_1}$
- $\mathbf{h}^{(i)} = \sigma\left(W^{(i)}\mathbf{h}^{(i-1)} + \mathbf{b}^{(i)}\right)$ for $W^{(i)} \in \mathbb{R}^{m_i \times m_{i-1}}, \mathbf{b}^{(i)} \in \mathbb{R}^{m_i}$ for $i = 2, \dots, L-1$
- $\mathcal{N}(\mathbf{x}) = \mathbf{h}^{(L)} = W^{(L)}\mathbf{h}^{(L-1)} + \mathbf{b}^{(L)}$ for $W^{(L)} \in \mathbb{R}^{1 \times m_{L-1}}, \mathbf{b}^{(L)} \in \mathbb{R}^1$

The *width* of the network is $\max\{m_1, \dots, m_{L-1}\}$. We define the *number of parameters* of the network as the number of weights of $\mathcal{N}$ which are non-zero. It corresponds to the number of edges when we view the neural network as a directed acyclic graph. We note that this definition is standard in the literature on VC dimension bounds for neural networks (cf. Bartlett et al. (2019)).

BIT COMPLEXITY

We refer to the *bit complexity of a weight* as the number of bits required to represent the weight. Throughout the paper we use only weights which have a finite bit complexity. Specifically, for $n \in \mathbb{N}$, its bit complexity is $\lceil \log n \rceil$. The *bit complexity of the network* is the maximal bit complexity of its weights.

INPUT DATA SAMPLES

In this work we assume that we are given $N$ labeled data samples $(\mathbf{x}_1, y_1), \dots, (\mathbf{x}_N, y_N) \in \mathbb{R}^d \times [C]$, and our goal is to construct a network $F : \mathbb{R}^d \to \mathbb{R}$ such that $F(\mathbf{x}_i) = y_i$ for every $i \in [N]$. We will assume that there is a separation parameter $\delta > 0$ such that for every $i \neq j$ we have $\|\mathbf{x}_i - \mathbf{x}_j\| \geq \delta$. We will also assume that the data samples have a bounded norm, i.e. there is $r > 0$ such that $\|\mathbf{x}_i\| \leq r$ for every $i \in [N]$.

Note that any neural network that does not ignore input neurons must have at least $d$ parameters. Existing memorization results (e.g., Park et al. (2020)) assume that $d$ is constant. As we discuss in Remark 3.2, in our work we assume that $d$ is at most $O(\sqrt{N})$. We also note that since our bounds depend logarithmically on $\delta$, then even if $\delta$ depends polynomially on $N$ it will not change our asymptotic bounds.

## 3   MEMORIZATION USING $\tilde{O}\left(\sqrt{N}\right)$ PARAMETERS

In this section we prove that given a finite set of $N$ labeled data points, there is a network with $\tilde{O}\left(\sqrt{N}\right)$ parameters which memorizes them. Formally, we have the following:

**Theorem 3.1.** *Let $N, d, C \in \mathbb{N}$, and $r, \delta > 0$, and let $(\mathbf{x}_1, y_1), \dots, (\mathbf{x}_N, y_N) \in \mathbb{R}^d \times [C]$ be a set of $N$ labeled samples with $\|\mathbf{x}_i\| \leq r$ for every $i$ and $\|\mathbf{x}_i - \mathbf{x}_j\| \geq \delta$ for every $i \neq j$. Denote $R := 10rN^2\delta^{-1}\sqrt{\pi d}$. Then, there exists a neural network $F : \mathbb{R}^d \to \mathbb{R}$ with width 12 and depth*

$$O\left(\sqrt{N \log(N)} + \sqrt{\frac{N}{\log(N)}} \cdot \max\{\log(R), \log(C)\}\right),$$

*such that $F(\mathbf{x}_i) = y_i$ for every $i \in [N]$.*

From the above theorem, we get that the total number of parameters is $\tilde{O}\left(d + \sqrt{N}\right)$. For $d = O(\sqrt{N})$ (see Remark 3.2 below) we get the lower bound presented in Theorem 1.1. Note that except for the number of data points $N$ and the dimension $d$, the number of parameters of the network depends only logarithmically on all the other parameters of the problem (namely, on $C, r, \delta^{-1}$). We also note that the construction can be improved by some constant factors, but we preferred simplicity

over minor improvements. Specifically, we hypothesize that it is possible to give a construction using width 3 (instead of 12) similar to the result in Park et al. (2020). To simplify the terms in the theorem, we assume that $r \geq 1$, otherwise (i.e., if all data points have a norm smaller than 1) we just fix $r := 1$ and get the same result. In the same manner, we assume that $\delta \leq 1$, otherwise we just fix $\delta := 1$.

**Remark 3.2** (Dependence on $d$). *Note that any neural network that does not ignore input neurons must have at least $d$ parameters. In our construction the first layer consists of a single neuron (i.e width 1), which means that the number of parameters in the first layer is $d + 1$. Hence, this dependence on $d$ is unavoidable. Previous works (e.g., Park et al. (2020)) assumed that $d$ is constant. In our work, to achieve the bound of $\tilde{O}\left(\sqrt{N}\right)$ parameters we can assume that either $d$ is constant or it may depend on $N$ with $d = O\left(\sqrt{N}\right)$.*

**Remark 3.3** (From classification to regression). *Although the theorem considers multi-class classification, it is possible to use the method suggested in Park et al. (2020) to extend the result to regression. Namely, if the output is in some bounded interval, then we can partition it into smaller intervals of length $\epsilon$ each. We define a set of output classes $C$ with $O\left(\frac{1}{\epsilon}\right)$ classes, such that each class corresponds to an interval. Now, we can use Theorem 3.1 to get an $\epsilon$ approximation of the output, while the number of parameters is linear in $\log\left(\frac{1}{\epsilon}\right)$.*

### 3.1 PROOF INTUITION

Below we describe the main ideas of the proof. The full proof can be found in Appendix A. The proof is divided into three stages, where at each stage we construct a network $F_i$ for $i \in \{1, 2, 3\}$, and the final network is $F = F_3 \circ F_2 \circ F_1$. Each subnetwork $F_i$ has width $O(1)$, but the depth varies for each such subnetwork.

**Stage I:** We project the data points from $\mathbb{R}^d$ to $\mathbb{R}$. This stage is similar to the one used in the proof of the main result from Park et al. (2020). We use a network with 2 layers for the projection. With the correct scaling, the output of this network on $\mathbf{x}_1, \ldots, \mathbf{x}_N \in \mathbb{R}^d$ are points $x_1, \ldots, x_N \in \mathbb{R}$ with $|x_i - x_j| \geq 2$ for $i \neq j$, and $|x_i| \leq R$ for every $i$ where $R = O\left(N^2 dr\delta^{-1}\right)$.

**Stage II:** Following the previous stage, our goal is now to memorize $(x_1, y_1), \ldots, (x_N, y_N) \in \mathbb{R} \times [C]$, where the $x_i$'s are separated by a distance of at least 2. We can also reorder the indices to assume w.l.o.g that $x_1 < x_2 < \cdots < x_N$. We split the data points into $\sqrt{N \log(N)}$ intervals, each containing $\sqrt{\frac{N}{\log(N)}}$ data points. We also construct crafted integers $w_1 \ldots, w_{\sqrt{N \log(N)}}$ and $u_1, \ldots, u_{\sqrt{N \log(N)}}$ in the following way: Note that by the previous stage, if we round $x_i$ to $\lfloor x_i \rfloor$, it can be represented using $\log(R)$ bits. Also, each $y_i$ can be represented by at most $\log(C)$ bits. Suppose that $x_i$ is the $k$-th data point in the $j$-th interval (where $j \in \left[\sqrt{N \log(N)}\right]$ and $k \in \left\{0, \ldots, \sqrt{\frac{N}{\log(N)}} - 1\right\}$), then we define $u_j, w_j$ such that

$$\text{BIN}_{k \cdot \log(R)+1:(k+1) \cdot \log(R)}(u_j) = \lfloor x_i \rfloor$$
$$\text{BIN}_{k \cdot \log(C)+1:(k+1) \cdot \log(C)}(w_j) = y_i \, .$$

That is, for each interval $j$, the number $u_j$ has $\log(R)$ bits which represent the integral value of the $k$-th data point in this interval. In the same manner, $w_j$ has $\log(C)$ bits which represent the label of the $k$-th data point in this interval. This is true for all the $k$ data points in the $j$-th interval, hence $w_j$ and $u_j$ are represented with $\log(R) \cdot \sqrt{\frac{N}{log(N)}}$ and $\log(C) \cdot \sqrt{\frac{N}{log(N)}}$ bits respectively.

We construct a network $F_2 : \mathbb{R} \to \mathbb{R}^3$ such that $F_2(x_i) = \begin{pmatrix} x_i \\ w_{j_i} \\ u_{j_i} \end{pmatrix}$ for each $x_i$, where the $i$-th data point is in the $j_i$-th interval. The integers $w_j$ and $u_j$ are used as parameters of the network $F_2$.

**Stage III:** In this stage we construct a network which uses a bit extraction technique adapting Telgarsky's triangle function (Telgarsky, 2016) to extract the relevant information out of the crafted

integers from the previous stage. In more details, given the input $\begin{pmatrix} x \\ w \\ u \end{pmatrix}$, we sequentially extract from $u$ the bits in places $k \cdot \log(R) + 1$ until $(k+1) \cdot \log(R)$ for $k \in \left\{ 0, 1, \ldots, \sqrt{\frac{N}{\log(N)}} - 1 \right\}$, and check whether $x$ is at distance at most 1 from the integer represented by those bits. If it is, then we extract the bits in places $k \cdot \log(C) + 1$ until $(k+1) \cdot \log(C)$ from $w$, and output the integer represented by those bits. By the previous stage, for each $x_i$ we know that $u$ includes the encoding of $\lfloor x_i \rfloor$, and since $|x_i - x_j| \geq 2$ for every $j \neq i$ (by the first stage), there is exactly one such $x_i$. Hence, the output of this network is the correct label for each $x_i$.

The construction is inspired by the works of Bartlett et al. (2019); Park et al. (2020). Specifically, the first stage uses similar results from Park et al. (2020) on projection onto a 1-dimensional space. The main difference from previous constructions is in stages II and III where in our construction we encode the input data points in the weights of the network. This is the main technique which allows us to reduce the required number of parameters from the $\tilde{O}\left(N^{2/3}\right)$ bound in Park et al. (2020) to $\tilde{O}\left(N^{1/2}\right)$. We also note, that although we use the convention of counting the number of parameters without zero-weights, in this section we do count zero weights, hence this result is comparable to Park et al. (2020) where they do count zero weights.

## 4 ON THE OPTIMAL NUMBER OF PARAMETERS

In section 3 we showed that a network with width $O(1)$ and depth $\tilde{O}\left(\sqrt{N}\right)$ can perfectly memorize $N$ data points, hence only $\tilde{O}\left(\sqrt{N}\right)$ parameters are required for memorization. In this section, we compare our results with the known lower bounds.

First, note that our problem contains additional parameters besides $N$. Namely, $d, C, r$ and $\delta$. As we discussed in Remark 3.2, we assume that $d$ is either constant or depends on $N$ with $d = O(\sqrt{N})$. In this comparison we also assume that $r$ and $\delta^{-1}$ are either constants or depend polynomially on $N$. We note that for a constant $d$, either $r$ or $\delta^{-1}$ must depend on $N$. For example, assume $d = 1$ and $r = 1/2$, then to have $N$ point in $\mathbb{R}^d$ with norm at most $r$ and distance at least $\delta$ from one another we must have that $\delta^{-1} \geq N - 1$. Hence, we can bound $R \leq \text{poly}(N)$ (where $R$ is defined in Theorem 3.1), which implies $\log(R) = O(\log(N))$. Moreover, we assume that $C \leq \text{poly}(N)$, because it is reasonable to expect that the number of output classes is not larger than the number of data points. Hence, also $\log(C) = O(\log(N))$. In regression tasks, as discussed in Remark 3.3, we can choose $C$ to consist of $\left\lceil \frac{1}{\epsilon} \right\rceil$ classes where $\epsilon$ is either a constant or depends at most polynomially on the other parameters.

Using that $R, C \leq \text{poly}(N)$, and tracing back the bounds given in Theorem 3.1, we get that the number of parameters of the network is $O\left(\sqrt{N \log(N)}\right)$. Note that here the $O(\cdot)$ notation only hides constant factors, which can also be exactly calculated using Theorem 3.1.

By Goldberg and Jerrum (1995), the VC dimension of the class of ReLU networks with $W$ parameters is $O(W^2)$. Thus, the maximal number of points that can be shattered is $O(W^2)$. Hence, if we can shatter $N$ points then $W = \Omega(\sqrt{N})$. In particular, it gives a $\Omega(\sqrt{N})$ lower bound for the number of parameters required to memorize $N$ inputs. Thus, the gap between our upper bound and the above lower bound is only $O\left(\sqrt{\log(N)}\right)$, which is sub-logarithmic.

Moreover, it implies that the number of parameters required to shatter one size-$N$ set is roughly equal (up to a sub-logarithmic factor) to the number of parameters required to memorize all size-$N$ sets (that satisfy some mild assumptions). Thus, perhaps surprisingly, the task of memorizing all size-$N$ sets is not significantly harder than shattering a single size-$N$ set.

If we further assume that $C$ depends on $N$, i.e., the number of classes depends on the number of samples (or in regression tasks, as we discussed in Remark 3.3, the accuracy depends on the number of samples), then we can show that our bound is tight up to *constant* terms. Thus, in this case the $\sqrt{\log(N)}$ factor is unavoidable. Formally, we have the following lemma:

**Lemma 4.1.** *Let $C, N \in \mathbb{N}$, and assume that $C = N^\epsilon$ for some constant $\epsilon > 0$. If we can express all the functions of the form $g : [N] \to [C]$ using neural networks with $W$ parameters, then $W = \Omega\left(\sqrt{N \log(N)}\right)$.*

*Proof.* Let $\mathcal{F}$ be the class of all the functions $f : [N] \times [\epsilon \log(N)] \to \{0, 1\}$. The VC-dimension bound from Goldberg and Jerrum (1995) implies that expressing all the functions from $\mathcal{F}$ with neural networks requires $W = \Omega\left(\sqrt{N \log(N)}\right)$ parameters, since it is equivalent to shattering a set of size $\epsilon N \log(N)$. Assume that we can express all the functions of the form $g : [N] \to [C]$ using networks with $W'$ parameters. Given some function $f \in \mathcal{F}$ we can express it with a neural network as follows: define a function $g : [N] \to [C]$ such that for every $n \in [N]$ and $i \in [\epsilon \log(N)]$, the $i$-th bit of $g(n)$ is $f(n, i)$. We construct a neural network for $f$, such that for an input $(n, i)$ it computes $g(n)$ (using $W'$ parameters) and outputs its $i$-th bit. The extraction of the $i$-th bit can be implemented using the bit extraction technique from the proof of Theorem 3.1. Overall, this network requires $O(W' + \log(N))$ parameters, and in this manner we can express all the functions in $\mathcal{F}$. This shows that $W = O(W' + \log(N))$, but since we also have $W = \Omega\left(\sqrt{N \log(N)}\right)$ then $W' = \Omega\left(\sqrt{N \log(N)}\right)$. $\square$

## 5 Limiting the Depth

In this section we generalize Theorem 3.1 to the case of having a bounded depth. We will then compare our upper bound on memorization to the VC-dimension bound from Bartlett et al. (2019). We have the following:

**Theorem 5.1.** *Assume the same setting as in Theorem 3.1, and denote $R := dNCr\delta^{-1}$. Let $L \in [\sqrt{N}]$. Then, there exists a network $F : \mathbb{R}^d \to \mathbb{R}$ with width $O\left(\frac{N}{L^2}\right)$, depth $O\left(\frac{L}{\sqrt{\log(L)}} \cdot \log(R)\right)$ and a total of $O\left(\frac{N}{L\sqrt{\log(L)}} \cdot \log(R) + d\right)$ parameters such that $F(\mathbf{x}_i) = y_i$ for every $i \in [N]$.*

Note that for $L = \sqrt{N}$ we get a similar bound to Theorem 3.1. In the proof we partition the dataset into $\frac{N}{L^2}$ subsets, each containing $L^2$ data points. We then use Theorem 3.1 on each such subset to construct a subnetwork of depth $O\left(L \cdot \log(R)\right)$ and width $O(1)$ to memorize the data points in the subset. We construct these subnetworks such that their output on each data point outside of their corresponding subset is zero. Finally we stack these subnetworks to construct a wide network, whose output is the sum of the outputs of the subnetworks.

By Bartlett et al. (2019), the VC dimension of the class of ReLU networks with $W$ parameters and depth $L$ is $O(WL \log(W))$. It implies that if we can shatter $N$ points with networks of depth $L$, then $W = \Omega\left(\frac{N}{L \log(N)}\right)$. As in section 4, assume that $d$ is either constant or at most $O(\sqrt{N})$, and that $r, \delta^{-1}$ and $C$ are bounded by some poly$(N)$. Theorem 5.1 implies that we can *memorize* any $N$ points (under the mild separability assumption) using networks of depth $L$ and $W = \tilde{O}\left(\frac{N}{L}\right)$ parameters. Therefore, the gap between our upper bound and the above lower bound is logarithmic. It also implies that, up to logarithmic factors, the task of memorizing any set of $N$ points using depth-$L$ neural networks is not more difficult than shattering a single set of $N$ points with depth-$L$ networks.

*Proof of Theorem 5.1.* We first construct a network $H : \mathbb{R}^d \to \mathbb{R}$ in the same manner as the construction of $F_1$ is the first stage of Theorem 3.1 (this construction is detailed in Lemma A.2 in the appendix). This is a 2-layer network with width 1. We denote the output of $H$ on the samples $\mathbf{x}_1, \ldots, \mathbf{x}_N$ as $x_1, \ldots, x_N$. Note that by the construction $|x_i| \leq O(R)$ for every $i \in [N]$ and $|x_i - x_j| \geq 2$ for every $i \neq j$.

We split the inputs to $\frac{N}{L^2}$ subsets of size $L^2$ each, we denote these subsets as $I_1, \ldots, I_{\frac{N}{L^2}}$ (assume w.l.o.g. that $\frac{N}{L^2}$ is an integer, otherwise replace it with $\left\lceil \frac{N}{L^2} \right\rceil$). For each subset $I_k$ we use stages II and III from Theorem 3.1 to construct a network $F_k$ such that $F_k(x_i) = y_i$ if $x_i \in I_k$. Note that we

only need to use the last two stages since we have already done the projection stage, hence are data samples are already one-dimensional. We construct a network $\tilde{F} : \mathbb{R} \to \mathbb{R}^{N/L^2}$ such that:

$$\tilde{F}(x) = \begin{pmatrix} F_1(x) \\ \vdots \\ F_{N/L^2}(x) \end{pmatrix} .$$

We also define a network $G : \mathbb{R}^{N/L^2} \to \mathbb{R}$ which adds up all its inputs. Finally, we construct the network $F : \mathbb{R}^d \to \mathbb{R}$ as $F := G \circ \tilde{F} \circ H$.

By the construction of each $F_k$ from Theorem 3.1, for every $x \in \mathbb{R}$, if $|x - x_i| \geq 2$ for every $i \in I_k$, then $F_k(x) = 0$. For every $i \in [N]$, there is exactly one $k \in [N/L^2]$ such that $i \in I_k$. Using the projection $H$, for this $k$ we get that $F_k(x_i) = y_i$, and for every $\ell \neq k$ we get that $F_\ell(x_i) = 0$. This means that $F(x_i) = y_i$ for every $i \in [N]$.

The depth of $F$ is the sum of the depths of its subnetwork. Both $H$ and $G$ have depth $O(1)$, and by Theorem 3.1, the depth of $F_k$ is at most $O\left( \frac{L}{\sqrt{\log(L)}} \cdot \log(R) \right)$, since each such network memorizes $L^2$ samples. Hence, the total depth of $F$ is $O\left( \frac{L}{\sqrt{\log(L)}} \cdot \log(R) \right)$. Finally, the width of $F$ is the maximal width of its subnetworks. The width of $G$ and $H$ is 1. The width of each $F_k$ is also $O(1)$ by Theorem 3.1, hence the width of $\tilde{F}$ is $O\left( \frac{N}{L^2} \right)$, which is also the width of $F$.

Recall that the number of parameters of a network is the number of non-zero weights. In our case, the network consists of $\frac{N}{L^2}$ subnetworks, where the weights between these subnetworks are zero. Each such subnetwork has depth $O\left( \frac{L}{\sqrt{\log(L)}} \cdot \log(R) \right)$ and width $O(1)$. Also, the projection phase requires $O(d)$ parameters. In total, the number of parameters in the network is $O\left( \frac{N}{L\sqrt{\log(L)}} \cdot \log(R) + d \right)$. □

## 6 BIT COMPLEXITY - LOWER AND UPPER BOUNDS

In the proof of Theorem 3.1 the bit complexity of the network is roughly $\sqrt{N \log(N)}$ (See Theorem A.1 in the appendix). On one hand, having such large bit complexity allows us to "store" and "extract" information from the weights of the network using bit extraction techniques. This enable us to memorize $N$ data points using significantly less than $N$ parameters. On the other hand, having large bit complexity makes the network difficult to implement on finite precision machines.

In this section we argue that having large bit complexity is *necessary* and *sufficient* if we want to construct a network that memorizes $N$ data points with less than $N$ parameters. We show both upper and lower bounds on the required bit complexity, and prove that, up to logarithmic factors, our construction is optimal w.r.t. the bit complexity.

First, we show a simple upper bound on the VC dimension of neural networks with bounded bit complexity. For this upper bound, we view the neural network as a directed graph with $U$ neurons, and where each weight is represented by $B$ bits. Note that representing each parameter requires $\log(B) + 2\log(U)$ bits, that is $\log(B)$ bits for the weights magnitude, and $2\log(U)$ bits for the input and output neuron of each edge.

**Theorem 6.1.** *Let $\mathcal{H}$ be the hypothesis class of ReLU neural networks with $W$ parameters, where each parameter is represented by $B$ bits. Then, the VC dimension of $\mathcal{H}$ is $O(WB + W\log(W))$.*

*Proof.* We claim that $\mathcal{H}$ is a finite hypothesis class with at most $2^{O(WB + W\log(W))}$ different functions. Let $f$ be a neural network in $\mathcal{H}$, and suppose that it has at most $W$ parameters and $U \leq W$ neurons. Then, each weight of $f$ can be represented by at most $B + 2\log(U)$ bits, namely, $2\log(U)$ bits for the indices of the input and output neurons of the edge, and $B$ bits for the weight magnitude. Hence, $f$ is represented by at most $O(W(B + \log(U)))$ bits. Since $U \leq W$ we get that every function in $\mathcal{H}$ can be represented by at most $O(W(B + \log(W)))$ bits, which gives the bound on the

size of $\mathcal{H}$. An upper bound on the VC dimension of finite classes is the log of their size (cf. Shalev-Shwartz and Ben-David (2014)), and hence the VC dimension of $\mathcal{H}$ is $O(WB + W\log(W))$. ☐

This lemma can be interpreted in the following way: Assume that we want to shatter a single set of $N$ points, and we use ReLU neural networks with $N^{1-\epsilon}$ parameters for some $\epsilon \in \left[0, \frac{1}{2}\right]$. Then, the bit complexity of each weight in the network must be at least $\Omega(N^\epsilon)$. Thus, to shatter $N$ points with less than $N$ parameters, we must have a neural network with bit complexity that depends on $N$. Also, for $\epsilon = \frac{1}{2}$, our construction in Theorem 3.1 (which memorizes any set of $N$ points) is optimal, up to logarithmic terms, since having bit complexity of $\Omega(\sqrt{N})$ is unavoidable. We emphasize that existing works that show non-trivial VC bounds also use large bit complexity (e.g. Bartlett et al. (1998; 2019)).

We now show an upper bound on the number of parameters of a network for memorizing $N$ data points, assuming that the bit complexity of the network is bounded:

**Theorem 6.2.** *Assume the same setting as in Theorem 3.1, and denote* $R := dNCr\delta^{-1}$. *Let* $B \in$ $\left[\sqrt{N}\right]$. *Then, there exists a network* $F : \mathbb{R}^d \to \mathbb{R}$ *with bit complexity* $O\left(\frac{B}{\sqrt{\log(B)}} \cdot \log(R)\right)$, *depth* $O\left(\frac{N\sqrt{\log(B)}}{B} \cdot \log(R)\right)$ *and width* $O(1)$ *such that* $F(\mathbf{x}_i) = y_i$ *for every* $i \in [N]$.

The full proof can be found in Appendix B. The proof idea is to partition the dataset into $\frac{N}{B^2}$ subsets, each containing $B^2$ data points. For each such subset we construct a subnetwork using Theorem 3.1 to memorize the points in this subset. We concatenate all these subnetwork to create one deep network, such that the output of each subnetwork is added to the output of the previous one. Using a specific projection from Lemma A.2 in the appendix, this enables each subnetwork to output 0 for each data point which is not in the corresponding subset. We get that the concatenated network successfully memorizes the $N$ given data points.

Assume, as in the previous sections, that $r, \delta^{-1}$ and $C$ are bounded by some poly($N$) and $d$ is bounded by $O(\sqrt{N})$. Theorem 6.2 implies that we can memorize any $N$ points (under mild assumptions), using networks with bit complexity $B$ and $\tilde{O}\left(\frac{N}{B}\right)$ parameters. More specifically, we can memorize $N$ points using networks with bit complexity $\tilde{O}\left(N^\epsilon\right)$ and $\tilde{O}\left(N^{1-\epsilon}\right)$ parameters, for $\epsilon \in \left[0, \frac{1}{2}\right]$. Up to logarithmic factors in $N$, this matches the bound implied by Theorem 6.1.

## 7 CONCLUSIONS

In this work we showed that memorization of $N$ separated points can be done using a feedforward ReLU neural network with $\tilde{O}\left(\sqrt{N}\right)$ parameters. We also showed that this construction is optimal up to logarithmic terms. This result is generalized for the cases of having a bounded depth network, and a network with bounded bit complexity. In both cases, our constructions are optimal up to logarithmic terms.

An interesting future direction is to understand the connection between our results and the optimization process of neural networks. In more details, it would be interesting to study whether training neural networks with standard algorithms (e.g. GD or SGD) can converge to a solution which memorizes $N$ data samples while the network have significantly less than $N$ parameters.

Another future direction is to study the connection between the bounds from this paper and the generalization capacity of neural networks. The double descent phenomenon (Belkin et al. (2019)) suggests that after a network crosses the "interpolation threshold", it is able to generalize well. Our results suggest that this threshold may be much smaller than $N$ for a dataset with $N$ samples, and it would be interesting to study if this is true also in practice.

Finally, it would be interesting to understand whether the logarithmic terms on our upper bounds are indeed necessary for the construction, or these are artifacts of our proof techniques. If these logarithmic terms are not necessary, then it will show that for neural network, the tasks of shattering a single set of $N$ points, and memorizing any set of $N$ points are exactly as difficult (maybe up to constant factors).

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

# A   PROOF FROM SECTION 3

We first give the proofs for each of the three stages of the construction, then we combine all the stages to prove Theorem 3.1. For convenience, we rewrite the theorem, and also state the bound on the bit complexity of the network:

**Theorem A.1.** *Let $N, d, C \in \mathbb{N}$, $r, \delta > 0$, and let $(\mathbf{x}_1, y_1), \ldots, (\mathbf{x}_N, y_N) \in \mathbb{R}^d \times [C]$ be a set of $N$ labeled samples with $\|\mathbf{x}_i\| \leq r$ for every $i$ and $\|\mathbf{x}_i - \mathbf{x}_j\| \geq \delta$ for every $i \neq j$. Denote $R := 10rN^2\delta^{-1}\sqrt{\pi d}$. Then, there exists a neural network $F : \mathbb{R}^d \to \mathbb{R}$ with width 12, depth $O\left(\sqrt{N \log(N)} + \sqrt{\frac{N}{\log(N)}} \cdot \max\{\log(R), \log(C)\}\right)$ and bit complexity bounded by $O\left(\log(d) + \sqrt{\frac{N}{\log(N)}} \cdot \max\{\log(R), \log(C)\}\right)$ such that $F(\mathbf{x}_i) = y_i$ for every $i \in [N]$.*

## A.1   STAGE I: PROJECTING ONTO A ONE-DIMENSIONAL SUBSPACE

**Lemma A.2.** *Let $\mathbf{x}_1, \ldots, \mathbf{x}_N \in \mathbb{R}^d$ with $\|\mathbf{x}_i\| \leq r$ for every $i$ and $\|x_i - x_j\| \geq \delta$ for every $i \neq j$. Then, there exists a neural network $F : \mathbb{R}^d \to \mathbb{R}$ with width 1, depth 2, and bit complexity $\log\left(3drN^2\sqrt{\pi}\delta^{-1}\right)$, such that $0 \leq F(\mathbf{x}_i) \leq 10rN^2\delta^{-1}\sqrt{\pi d}$ for every $i \in [N]$ and $|F(\mathbf{x}_i) - F(\mathbf{x}_j)| \geq 2$ for every $i \neq j$.*

To show this lemma we use a similar argument to the projection phase from Park et al. (2020), where the main difference is that we use weights with bounded bit complexity. Specifically, we use the following:

**Lemma A.3** (Lemma 8 from Park et al. (2020))**.** *Let $N, d \in \mathbb{N}$, then for any distinct $\mathbf{x}_1, \ldots, \mathbf{x}_N \in \mathbb{R}^d$ there exists a unit vector $\mathbf{u} \in \mathbb{R}^d$ such that for all $i \neq j$:*

$$\sqrt{\frac{8}{\pi d}} \cdot \frac{1}{N^2} \|\mathbf{x}_i - \mathbf{x}_j\| \leq |\mathbf{u}^\top (\mathbf{x}_i - \mathbf{x}_j)| \leq \|\mathbf{x}_i - \mathbf{x}_j\| \tag{1}$$

*Proof of Lemma A.2.* We first use Lemma A.3 to find $\mathbf{u} \in \mathbb{R}^d$ that satisfies Eq. (1). Note that $\|\mathbf{u}\| = 1$, hence every coordinate of $\mathbf{u}$ is smaller than 1. We define $\tilde{\mathbf{u}} \in \mathbb{R}^d$ such that each of its coordinates is equal to the first $\lceil \log\left(dN^2\sqrt{\pi}\right) \rceil$ bits of the corresponding coordinate of $\mathbf{u}$. Note that $\|\tilde{\mathbf{u}} - \mathbf{u}\| \leq \frac{\sqrt{d}}{2^{\log(dN^2\sqrt{\pi})}} \leq \frac{1}{N^2\sqrt{\pi d}}$. For every $i \neq j$ we have that:

$$|\tilde{\mathbf{u}}^\top(\mathbf{x}_i - \mathbf{x}_j)| \geq |\mathbf{u}^\top(\mathbf{x}_i - \mathbf{x}_j)| - |(\tilde{\mathbf{u}} - \mathbf{u})^\top(\mathbf{x}_i - \mathbf{x}_j)|$$

$$\geq \sqrt{\frac{8}{\pi d}} \frac{1}{N^2} \|\mathbf{x}_i - \mathbf{x}_j\| - \|\tilde{\mathbf{u}} - \mathbf{u}\| \cdot \|\mathbf{x}_i - \mathbf{x}_j\|$$

$$\geq \sqrt{\frac{8}{\pi d}} \frac{1}{N^2} \|\mathbf{x}_i - \mathbf{x}_j\| - \frac{1}{N^2\sqrt{\pi d}} \cdot \|\mathbf{x}_i - \mathbf{x}_j\|$$

$$\geq \frac{1}{N^2\sqrt{\pi d}} \|\mathbf{x}_i - \mathbf{x}_j\| . \tag{2}$$

We also have for every $i \in [N]$:

$$|\tilde{\mathbf{u}}^\top \mathbf{x}_i| \leq |\mathbf{u}^\top \mathbf{x}_i| + |(\tilde{\mathbf{u}} - \mathbf{u})^\top \mathbf{x}_i|$$

$$\leq \|\mathbf{x}_i\| + \frac{1}{N^2\sqrt{\pi d}} \|\mathbf{x}_i\| \leq 2\|\mathbf{x}_i\| . \tag{3}$$

Let $b := -\min\{0, \min_i\{\lfloor \tilde{\mathbf{u}}^\top \mathbf{x}_i \rfloor\}\} + 1$, and note that by Eq. (3) we have $b \leq 2r + 1$. We define the network $F : \mathbb{R}^d \to \mathbb{R}$ in the following way:

$$F(\mathbf{x}) = (2N^2\delta^{-1}\sqrt{\pi d}) \cdot \sigma\left(\tilde{\mathbf{u}}^\top \mathbf{x} + b\right) .$$

We show the correctness of the construction. Let $i \neq j$, then we have that:

$$|F(\mathbf{x}_i) - F(\mathbf{x}_j)| = (2N^2\delta^{-1}\sqrt{\pi d})\left|\sigma\left(\tilde{\mathbf{u}}^\top \mathbf{x}_i + b\right) - \sigma\left(\tilde{\mathbf{u}}^\top \mathbf{x}_j + b\right)\right|$$

$$= (2N^2\delta^{-1}\sqrt{\pi d})\left|\tilde{\mathbf{u}}^\top \mathbf{x}_i + b - (\tilde{\mathbf{u}}^\top \mathbf{x}_j + b)\right|$$

$$= (2N^2\delta^{-1}\sqrt{\pi d})\left|\tilde{\mathbf{u}}^\top(\mathbf{x}_i - \mathbf{x}_j)\right| \geq 2 ,$$

where the second equality is because by the definition of $b$ we have that $\tilde{\mathbf{u}}^\top \mathbf{x}_i + b \geq 0$ for every $i$, and the last inequality is by Eq. (2) and the assumption that $\|\mathbf{x}_i - \mathbf{x}_j\| \geq \delta$ for every $i \neq j$. Now, let $i \in [N]$, then we have:

$$
\begin{aligned}
|F(\mathbf{x}_i)| &= (2N^2\delta^{-1}\sqrt{\pi d}) \cdot \sigma(\tilde{\mathbf{u}}^\top \mathbf{x}_i + b) \\
&= (2N^2\delta^{-1}\sqrt{\pi d}) \cdot (\tilde{\mathbf{u}}^\top \mathbf{x}_i + b) \\
&\leq (2N^2\delta^{-1}\sqrt{\pi d}) \cdot (4r + 1) \leq 10rN^2\delta^{-1}\sqrt{\pi d} \,,
\end{aligned}
$$

where the second to last inequality is since $b \leq 2r + 1$ and Eq. (3).

The network $F$ has depth 2 and width 1. The bit complexity of the network can be bounded by $\log\left(3drN^2\sqrt{\pi}\delta^{-1}\right)$. This is because each coordinate of $\tilde{\mathbf{u}}$ can be represented using $\lceil \log(dN^2\sqrt{\pi}) \rceil$ bits, the bias $b$ can be represented using at most $\lceil \log(2r+1) \rceil$ bits and the weight in the second layer can be represented using $\lceil \log(2N^2\sqrt{\pi d}\delta^{-1}) \rceil$ bits. □

## A.2 Stage II: Finding the right subset

**Lemma A.4.** *Let $x_1 < \cdots < x_N < R$ with $R > 0$, $x_i \in \mathbb{R}$ for every $i \in [N]$ and $|x_i - x_j| \geq 2$ for every $i \neq j$. Let $m \in \mathbb{N}$ with $m < N$ and let $w_1, \ldots, w_m \in \mathbb{N}$ where $\textsc{len}(w_j) \leq b$ for every $j \in [m]$. Let $k := \lceil \frac{N}{m} \rceil$. Then, there exists a neural network $F : \mathbb{R} \to \mathbb{R}$ with width 4, depth $3m + 2$ and bit complexity $b + \lceil \log(R) \rceil$ such that for every $i \in [N]$ we have that $F(x_i) = w_{\lceil \frac{i}{k} \rceil}$.*

*Proof.* Let $j \in [m]$. We define network blocks $\tilde{F}_j : \mathbb{R} \to \mathbb{R}$ and $F_j : \mathbb{R}^2 \to \mathbb{R}^2$ in the following way: First, we use Lemma A.6 to construct $\tilde{F}_j$ such that $\tilde{F}_j(x) = 1$ for every $x \in [\lfloor x_{j\cdot k - k + 1} \rfloor, \lfloor x_{j \cdot k} + 1 \rfloor]$, and $\tilde{F}_j(x) = 0$ for $x < \lfloor x_{j\cdot k - k + 1} \rfloor - \frac{1}{2}$ or $x > \lfloor x_{j\cdot k} + 1 \rfloor + \frac{1}{2}$. In particular, $\tilde{F}_j(x_i) = 1$ if $i \in [j \cdot k - k + 1, j \cdot k]$, and $\tilde{F}_j(x_i) = 0$ otherwise. Note that since $k$ is defined with ceil, it is possible that $j \cdot k \geq N$, if this is the case we replace $j \cdot k$ with $N$. Next, we define:

$$
F_j\left(\begin{pmatrix} x \\ y \end{pmatrix}\right) = \begin{pmatrix} x \\ y + w_j \cdot \tilde{F}_j(x) \end{pmatrix} \,.
$$

Finally we define the network $F(x) = \begin{pmatrix} 0 \\ 1 \end{pmatrix}^\top F_m \circ \cdots \circ F_1\left(\begin{pmatrix} x \\ 0 \end{pmatrix}\right)$ (we can use one extra layer to augment the input with an extra coordinate of 0).

We show that this construction is correct. Note that for every $i \in [N]$, if we denote $j = \lceil \frac{i}{k} \rceil$, then $\tilde{F}_j(x_i) = 1$ and for every $j' \neq j$ we have $\tilde{F}_{j'}(x_i) = 0$. By the construction of $F$ we get that $F(x_i) = w_{\lceil \frac{i}{k} \rceil}$.

The width of $F$ at every layer $j$ is at most the width of $\tilde{F}_j + 2$, since we also keep copies of both $x$ and $y$, hence the width is at most 4. The depth of $\tilde{F}_j$ is 2, and $F$ is a composition of the $\tilde{F}_j$'s with an extra layer for the input to get $x \mapsto \begin{pmatrix} x \\ 0 \end{pmatrix}$, and an extra layer in the output to extract the last coordinate. Hence, its depth is $3m + 2$. The bit complexity is bounded by the sum of the bit complexity of $\tilde{F}_j$ and the weights $w_j$, hence it is bounded by $b + \lceil \log(R) \rceil$. □

## A.3 Stage III: Bit extraction from the Crafted Weights

**Lemma A.5.** *Let $\rho, n, c \in \mathbb{N}$. Let $u \in \mathbb{N}$ with $\textsc{len}(u) = \rho \cdot n$ and let $w \in \mathbb{N}$ with $\textsc{len}(w) = c \cdot n$. Assume that for any $\ell, k \in \{0, 1, \ldots, n - 1\}$ with $\ell \neq k$ we have that $\left| \textsc{bin}_{\rho\cdot\ell+1:\rho\cdot(\ell+1)}(u) - \textsc{bin}_{\rho\cdot k+1:\rho\cdot(k+1)}(u) \right| \geq 2$. Then, there exists a network $F : \mathbb{R}^3 \to \mathbb{R}$ with width 12, depth $3n \cdot \max\{\rho, c\} + 2n + 2$ and bit complexity $n \max\{\rho, c\} + 2$, such that for every $x > 0$, if there exist $j \in \{0, 1, \ldots, n - 1\}$ where $\lfloor x \rfloor = \textsc{bin}_{\rho\cdot j+1:\rho\cdot(j+1)}(u)$, then:*

$$
F\left(\begin{pmatrix} x \\ w \\ u \end{pmatrix}\right) = \textsc{bin}_{c\cdot j+1:c\cdot(j+1)}(w) \,.
$$

*Proof.* Let $i \in \{0, 1, \ldots, n-1\}$, and denote by $\varphi(z) := \sigma(\sigma(2z) - \sigma(4z - 2))$ the triangle function due to Telgarsky (2016). We construct the following neural network $F_i$:

$$
F_i : \begin{pmatrix} x \\ \varphi^{(i\cdot\rho)}\left(\frac{u}{2^{n\cdot\rho}} + \frac{1}{2^{n\cdot\rho+1}}\right) \\ \varphi^{(i\cdot\rho)}\left(\frac{u}{2^{n\cdot\rho}} + \frac{1}{2^{n\cdot\rho+2}}\right) \\ \varphi^{(i\cdot c)}\left(\frac{w}{2^{n\cdot c}} + \frac{1}{2^{n\cdot c+1}}\right) \\ \varphi^{(i\cdot c)}\left(\frac{w}{2^{n\cdot c}} + \frac{1}{2^{n\cdot c+2}}\right) \\ y \end{pmatrix} \mapsto \begin{pmatrix} x \\ \varphi^{((i+1)\cdot\rho)}\left(\frac{u}{2^{n\cdot\rho}} + \frac{1}{2^{n\cdot\rho+1}}\right) \\ \varphi^{((i+1)\cdot\rho)}\left(\frac{u}{2^{n\cdot\rho}} + \frac{1}{2^{n\cdot\rho+2}}\right) \\ \varphi^{((i+1)\cdot c)}\left(\frac{w}{2^{n\cdot c}} + \frac{1}{2^{n\cdot c+1}}\right) \\ \varphi^{((i+1)\cdot c)}\left(\frac{w}{2^{n\cdot c}} + \frac{1}{2^{n\cdot c+2}}\right) \\ y + y_i \end{pmatrix}
$$

where we define $y_i := \text{BIN}_{i\cdot c+1:(i+1)\cdot c}(w)$ if $x \in [\text{BIN}_{i\cdot\rho+1:(i+1)\cdot\rho}(u), \ \text{BIN}_{i\cdot\rho+1:(i+1)\cdot\rho}(u)+1]$, and $y_i = 0$ if $x > \text{BIN}_{i\cdot\rho+1:(i+1)\cdot\rho}(u) + \frac{3}{2}$ or $x < \text{BIN}_{i\cdot\rho+1:(i+1)\cdot\rho}(u) - \frac{1}{2}$.

The construction uses two basic building blocks: The first is using Lemma A.7 twice, for $u$ and $w$. This way we construct two smaller networks $F_i^w, F_i^u$, such that:

$$
F_i^u : \begin{pmatrix} \varphi^{(i\cdot\rho)}\left(\frac{u}{2^{n\cdot\rho}} + \frac{1}{2^{n\cdot\rho+1}}\right) \\ \varphi^{(i\cdot\rho)}\left(\frac{u}{2^{n\cdot\rho}} + \frac{1}{2^{n\cdot\rho+2}}\right) \end{pmatrix} \mapsto \begin{pmatrix} \varphi^{((i+1)\cdot\rho)}\left(\frac{u}{2^{n\cdot\rho}} + \frac{1}{2^{n\cdot\rho+1}}\right) \\ \varphi^{((i+1)\cdot\rho)}\left(\frac{u}{2^{n\cdot\rho}} + \frac{1}{2^{n\cdot\rho+2}}\right) \\ \text{BIN}_{i\cdot\rho+1:(i+1)\cdot\rho}(u) \end{pmatrix}
$$

$$
F_i^w : \begin{pmatrix} \varphi^{(i\cdot c)}\left(\frac{w}{2^{n\cdot c}} + \frac{1}{2^{n\cdot c+1}}\right) \\ \varphi^{(i\cdot c)}\left(\frac{w}{2^{n\cdot c}} + \frac{1}{2^{n\cdot c+2}}\right) \end{pmatrix} \mapsto \begin{pmatrix} \varphi^{((i+1)\cdot c)}\left(\frac{w}{2^{n\cdot c}} + \frac{1}{2^{n\cdot c+1}}\right) \\ \varphi^{((i+1)\cdot c)}\left(\frac{w}{2^{n\cdot c}} + \frac{1}{2^{n\cdot c+2}}\right) \\ \text{BIN}_{i\cdot c+1:(i+1)\cdot c}(w) \end{pmatrix} .
$$

The second is to construct $y_i$. We use Lemma A.8 with inputs $x$ and $\text{BIN}_{i\cdot\rho+1:(i+1)\cdot\rho}(u)$, and denote the output of this network by $\tilde{y}_i$. We use the following 1-layer network:

$$
\begin{pmatrix} \tilde{y}_i \\ \text{BIN}_{i\cdot c+1:(i+1)\cdot c}(w) \end{pmatrix} \mapsto \sigma\left(\tilde{y}_i \cdot 2^{c+1} - 2^{c+1} + \text{BIN}_{i\cdot c+1:(i+1)\cdot c}(w)\right)
$$

Note that if $\tilde{y}_i = 1$ then the output of the above network is $\text{BIN}_{i\cdot c+1:(i+1)\cdot c}(w)$, and if $\tilde{y}_i = 0$ then the output of the network is 0 since $\text{BIN}_{i\cdot c+1:(i+1)\cdot c}(w) \leq 2^c$.

The last layer of $F_i$ is just adding the output $y_i$ to $y$, and using the identity on the other coordinates which are being kept as the output, while the other coordinates (namely, the last output coordinates of $F_i^w$ and $F_i^u$) do not appear in the output of $F_i$. Also note that all the inputs and outputs of the networks and basic building blocks are positive, hence $\sigma$ acts as the identity on them. This means that if $F_i^u$ is deeper than $F_i^w$ (or the other way around), then we can just add identity layers to $F_i^w$ which do not change the output, but make the depths of both networks equal. Finally we define:

$$
F := G \circ F_{n-1} \circ \cdots \circ F_0 \circ H ,
$$

where: (1) $G : \mathbb{R}^5 \to \mathbb{R}$ is a 1-layer network that outputs the last coordinate of the input, and; (2) $H : \mathbb{R}^3 \to \mathbb{R}^6$ is a 1-layer network such that:

$$
H : \begin{pmatrix} x \\ w \\ u \end{pmatrix} \mapsto \begin{pmatrix} x \\ \frac{u}{2^{n\cdot\rho}} + \frac{1}{2^{n\cdot\rho+1}} \\ \frac{u}{2^{n\cdot\rho}} + \frac{1}{2^{n\cdot\rho+2}} \\ \frac{w}{2^{n\cdot c}} + \frac{1}{2^{n\cdot c+1}} \\ \frac{w}{2^{n\cdot c}} + \frac{1}{2^{n\cdot c+2}} \\ 0 \end{pmatrix}
$$

where we assume that $x, w, u > 0$.

We show the correctness of the construction. We have that $F\left(\begin{pmatrix} x \\ w \\ u \end{pmatrix}\right) = \sum_{i=0}^{n-1} y_i$. Assume there exists $j \in \{0, 1, \ldots, n-1\}$ such that $\lfloor x \rfloor = \text{BIN}_{\rho\cdot j+1:\rho\cdot(j+1)}(u)$. Then, by the construction of $y_i$ we have that $y_j = \text{BIN}_{c\cdot j+1:c\cdot(j+1)}(w)$, and for every other $\ell \neq j$, since $\left|\text{BIN}_{\rho\cdot\ell+1:\rho\cdot(\ell+1)}(u) - \text{BIN}_{\rho\cdot j+1:\rho\cdot(j+1)}(u)\right| \geq 2$, we must have that $x > \text{BIN}_{\rho\cdot\ell+1:\rho\cdot(\ell+1)}(u) + \frac{3}{2}$ or $x < \text{BIN}_{\rho\cdot\ell+1:\rho\cdot(\ell+1)}(u) - \frac{1}{2}$, which means that $y_\ell = 0$. In total we get the the output of $F$ is equal to $\sum_{i=0}^{n-1} y_i = y_j = \text{BIN}_{c\cdot j+1:c\cdot(j+1)}(w)$ as required.

We will now calculate the width, depth and bit complexity of the network $F$. The width of each $F_i^w$ and $F_i^u$ is equal to 5 by Lemma A.7, and we need two more neurons for $x$ and $y$. In total, the width is bounded by 12. Note that all other parts of the network (i.e. $H$, $G$ and the network from Lemma A.8) require less width than this bound. For the depth, by Lemma A.7, the depth of each $F_i^u$ and $F_i^w$ is at most $3 \cdot \max\{\rho, c\}$, adding the construction from Lemma A.8, the depth of each $F_i$ is at most $3 \cdot \max\{\rho, c\} + 2$. Summing over all $i$, and adding the depth of $G$ and $H$ we get that the depth of $F$ is bounded by $3n \cdot \max\{\rho, c\} + 2n + 2$. Finally, the bit complexity of $F_i^u$, $F_i^w$ and $H$ is bounded by $n \max\{\rho, c\} + 2$, and all other parts of the network require less bit complexity. Hence, the bit complexity of $F$ is bounded by $n \cdot \max\{\rho, c\} + 2$. □

## A.4 PROOF OF THEOREM 3.1

The construction is done in three phases. We first project the points onto a 1-dimensional subspace, where the projection preserves distances up to some error. The second step is to split the points into $\sqrt{N \log(N)}$ subsets, and to extract two weights containing "hints" about the points and their labels. The third step is to parse the hints using an efficient bit extraction method, and to output the correct label for each point.

Throughout the proof we assume w.l.o.g. that the following terms are integers since we use them as indices: $\sqrt{N \log(N)}$, $\sqrt{\frac{N}{\log(N)}}$, $\log(R)$, $\log(C)$. If any of them is not an integer we can just replace it with its ceil (i.e. $\lceil \log(R) \rceil$ instead of $\log(R)$). This replacement changes these numbers by at most 1, which in turn can only increase the number of parameters or bit complexity by at most a constant factor. Since we give the result using the $O(\cdot)$ notation, this does not change it. Also, note that the width of the network is independent of such terms.

For the first stage, we use Lemma A.2 to construct a network $F_1 : \mathbb{R}^d \to \mathbb{R}$ such that $F_1(\mathbf{x}_i) \leq 10rN^2\delta^{-1}\sqrt{\pi d}$ for every $i \in [N]$ and $|F_1(\mathbf{x}_i) - F_1(\mathbf{x}_j)| \geq 2$ for every $i \neq j$.

We denote $R := 10rN^2\delta^{-1}\sqrt{\pi d}$, and we denote the output of $F_1$ on the samples $\mathbf{x}_1, \ldots, \mathbf{x}_N \in \mathbb{R}^d$ as $x_1, \ldots, x_N \in \mathbb{R}$ for simplicity. We also assume that the $x_i$'s are in increasing order, otherwise we reorder the indices. This concludes the first stage.

For the second stage, we define two sets of integers $w_1, \ldots, w_{\sqrt{N \log(N)}}$ each represented by $\sqrt{\frac{N}{\log(N)}} \cdot \log(C)$ bits, and $u_1, \ldots, u_{\sqrt{N \log(N)}}$ each represented by $\sqrt{\frac{N}{\log(N)}} \cdot \log(R)$ bits, in the following way: For every $i \in [N]$ let $j := \left\lceil i \cdot \sqrt{\frac{\log(N)}{N}} \right\rceil$ and $k := i \left( \bmod \sqrt{\frac{N}{\log(N)}} \right)$. We set:

$$\text{BIN}_{k \cdot \log(C)+1:(k+1) \cdot \log(C)}(w_j) = y_i$$
$$\text{BIN}_{k \cdot \log(R)+1:(k+1) \cdot \log(R)}(u_j) = \lfloor x_i \rfloor .$$

We now use Lemma A.4 twice to construct two networks $F_2^w : \mathbb{R} \to \mathbb{R}$ and $F_2^u : \mathbb{R} \to \mathbb{R}$ such that $F_2^w(x_i) = w_{j_i}$ and $F_2^u(x_i) = u_{j_i}$ for $j_i = \left\lceil i \cdot \sqrt{\frac{\log(N)}{N}} \right\rceil$. We construct the network $F_2 : \mathbb{R} \to \mathbb{R}$ to be a concatenation of the two networks, with an additional coordinate which outputs $\sigma(x)$ (since the inputs $x_i$ are positive, this coordinate just output the exact input). Namely, we construct a network such that for every $i \in [N]$ we have

$$F_2(x_i) = \begin{pmatrix} x_i \\ w_{j_i} \\ u_{j_i} \end{pmatrix} ,$$

where $j_i = \left\lceil i \cdot \sqrt{\frac{\log(N)}{N}} \right\rceil$.

For the third stage, we use Lemma A.5 to construct a network $F_3 : \mathbb{R}^3 \to \mathbb{R}$ such that for every $x > 0$, if there exist $j \in \left\{ 0, 1, \ldots, \left\lceil \sqrt{\frac{\log(N)}{N}} \right\rceil - 1 \right\}$ such that $\lfloor x \rfloor = \text{BIN}_{\log(R) \cdot j+1:\log(R) \cdot (j+1)}(u)$, then:

$$F_3 \left( \begin{pmatrix} x \\ w \\ u \end{pmatrix} \right) = \text{BIN}_{\log(C) \cdot j+1:\log(C) \cdot (j+1)}(w) .$$

Finally, we construct the network $F : \mathbb{R}^d \to \mathbb{R}$ as $F(\mathbf{x}) = F_3 \circ F_2 \circ F_1(\mathbf{x})$.

We show the correctness of the construction. Let $i \in [N]$, and let $y := F(\mathbf{x}_i)$. By the construction of $F_2$ and the numbers $w_1, \ldots, w_{\sqrt{N \log(N)}}, u_1, \ldots, u_{\sqrt{N \log(N)}}$, if we denote $j :=$ $\left\lceil i \cdot \sqrt{\frac{\log(N)}{N}} \right\rceil$ and $k := i \left( \bmod \sqrt{\frac{N}{\log(N)}} \right)$, then we have that: (1) $F_2 \circ F_1(\mathbf{x}_i) = \begin{pmatrix} F_1(\mathbf{x}_i) \\ w_j \\ u_j \end{pmatrix}$; (2) $\mathrm{BIN}_{\log(R) \cdot k + 1 : \log(R) \cdot (k+1)}(u_j) = \lfloor F_1(\mathbf{x}_i) \rfloor$; and (3) $\mathrm{BIN}_{\log(C) \cdot k + 1 : \log(C) \cdot (k+1)}(w_j) = y_i$. Finally, by the construction of $F_3$ we get that $y = F_3 \circ F_2 \circ F_1(\mathbf{x}_i) = \mathrm{BIN}_{\log(C) \cdot k + 1 : \log(C) \cdot (k+1)}(w_j) = y_i$ as required.

The width of the network $F$, namely the maximal width of its subnetworks, is the width of $F_3$, which is 12. The depth of $F$ is the sum of the depths of each of its subnetworks. The depth of $F_1$ is 2, the depth of $F_2$ is $O\left(\sqrt{N \log(N)}\right)$ and the depth of $F_3$ is $O\left(\sqrt{\frac{N}{\log(N)}} \cdot \max\{\log(R), \log(C)\}\right)$. Hence, the total depth of $F$ can be bounded by $O\left(\sqrt{N \log(N)} + \sqrt{\frac{N}{\log(N)}} \cdot \max\{\log(R), \log(C)\}\right)$.

The bit complexity of $F$ is the maximal bit complexity of its subnetworks. The bit complexity of $F_1$ is $\log\left(3drN^2\sqrt{\pi}\delta^{-1}\right) = \log(d) + \log(R/3)$, the bit complexity of $F_2$ is $O\left(\sqrt{\frac{N}{\log(N)}} \cdot \max\{\log(R), \log(C)\}\right)$ and the bit complexity of $F_3$ is also $O\left(\sqrt{\frac{N}{\log(N)}} \cdot \max\{\log(R), \log(C)\}\right)$. In total, the bit complexity of $F$ can be bounded by $O\left(\log(d) + \sqrt{\frac{N}{\log(N)}} \cdot \max\{\log(R), \log(C)\}\right)$.

### A.5 Auxiliary Lemmas

**Lemma A.6.** *Let $a, b \in \mathbb{N}$ with $a < b$. Then, there exists a neural network $F$ with depth 2, width 2 and bit complexity $\mathrm{LEN}(b)$ such that $F(x) = 1$ for $x \in [a, b]$ and $F(x) = 0$ for $x > b + \frac{1}{2}$ or $x < a - \frac{1}{2}$.*

*Proof.* Consider the following neural network:

$$F(x) = \sigma(1 - \sigma(2a - 2x)) + \sigma(1 - \sigma(2x - 2b)) - 1 \,.$$

It is easy to see that this networks satisfies the requirements. Also, its bit complexity is at most $\mathrm{LEN}(b)$, since $a < b$, hence $a$ can be represented by at most $\mathrm{LEN}(b)$ bits. $\qquad\square$

**Lemma A.7.** *Let $n \in \mathbb{N}$ and let $i, j \in \mathbb{N}$ with $i < j \leq n$. Denote Telgarsky's triangle function by $\varphi(z) := \sigma(\sigma(2z) - \sigma(4z - 2))$. Then, there exists a neural network $F : \mathbb{R}^2 \to \mathbb{R}^3$ with width 5, depth $3(j - i + 1)$, and bit complexity $n + 2$, such that for any $x \in \mathbb{N}$ with $\mathrm{LEN}(x) \leq n$, if the input of $F$ is $\begin{pmatrix} \varphi^{(i-1)}\left(\frac{x}{2^n} + \frac{1}{2^{n+1}}\right) \\ \varphi^{(i-1)}\left(\frac{x}{2^n} + \frac{1}{2^{n+2}}\right) \end{pmatrix}$, then it outputs: $\begin{pmatrix} \varphi^{(j)}\left(\frac{x}{2^n} + \frac{1}{2^{n+1}}\right) \\ \varphi^{(j)}\left(\frac{x}{2^n} + \frac{1}{2^{n+2}}\right) \\ \mathrm{BIN}_{i:j}(x) \end{pmatrix}$.*

*Proof.* We use Telgarsky's function to extract bits. Let $x \in \mathbb{N}$ with $\mathrm{LEN}(x) = n$, and let $i \in \mathbb{N}$ with $i \leq n$. Then, we have that:

$$\mathrm{BIN}_i(x) = 2^{n+2-i}\sigma\left(\varphi^{(i)}\left(\frac{x}{2^n} + \frac{1}{2^{n+2}}\right) - \varphi^{(i)}\left(\frac{x}{2^n} + \frac{1}{2^{n+1}}\right)\right) \,. \tag{4}$$

The intuition behind Eq. (4) is the following: The function $\varphi^{(i)}$ is a piecewise linear function with $2^{i-1}$ "bumps". Each such "bump" consists of two linear parts with a slope of $2^i$, the first linear part goes from 0 to 1, and the second goes from 1 to 0. Let $x \in \mathbb{N}$ with at most $n$ bits in its binary representation. It can be seen that the $i$-th bit of $x$ is 1 if $\varphi^{(i)}\left(\frac{x}{2^n} + \frac{1}{2^{n+1}}\right)$ is on the second linear part (i.e. descending from 1 to 0) and its $i$-th bit is 0 otherwise. This shows that $\varphi^{(i)}\left(\frac{x}{2^n} + \frac{1}{2^{n+2}}\right) - \varphi^{(i)}\left(\frac{x}{2^n} + \frac{1}{2^{n+1}}\right)$ is equal to $2^{i-n-2}$ if the $i$-th bit of $x$ is 1, and this expression is negative otherwise. The correctness of Eq. (4) follows.

Let $j, i \in \mathbb{N}$ with $i < j$ and denote $c := j - i$. Using the construction above as a building block, we construct a network which outputs $\text{BIN}_{i:j}(x)$ in the following way: For $\ell \in \{0, 1, \dots, c\}$ define $F_\ell : \mathbb{R}^3 \to \mathbb{R}^3$ to be the neural network, such that for an input $\begin{pmatrix} \varphi^{(i-1+\ell)}\left(\frac{x}{2^n} + \frac{1}{2^{n+1}}\right) \\ \varphi^{(i-1+\ell)}\left(\frac{x}{2^n} + \frac{1}{2^{n+2}}\right) \\ y \end{pmatrix}$, it outputs

$\begin{pmatrix} \varphi^{(i+\ell)}\left(\frac{x}{2^n} + \frac{1}{2^{n+1}}\right) \\ \varphi^{(i+\ell)}\left(\frac{x}{2^n} + \frac{1}{2^{n+2}}\right) \\ y + 2^{c-\ell}\text{BIN}_{i+\ell}(x) \end{pmatrix}$. The network $F_\ell$ is obtained by composing the first two coordinates with Telgarsky's function $\varphi$, and then the last coordinate is calculated using Eq. (4). Finally, we define: $F := F_c \circ \cdots \circ F_0$, and we augment the input of $F_0$, such that its last coordinate is zero. The output of $F$ is as required since

$$\sum_{\ell=0}^{c} 2^{c-\ell}\text{BIN}_{i+\ell}(x) = \text{BIN}_{i:i+c}(x) = \text{BIN}_{i:j}(x) .$$

Finally, we compute the width, depth and bit complexity of $F$. Its width is bounded by twice the width of $\varphi$ (which is 2), plus an extra neuron for computing the output $y$, hence its width is bounded by 5. Its depth is equal to $3(j - i + 1)$, since we need two layers to compute $\varphi$ and an extra layer to compute the output $y$. Finally, the bit complexity is bounded by $n + 2$, since the largest weight in the network is at most $2^{n+2}$. $\qquad \square$

**Lemma A.8.** *There exists a network $F : \mathbb{R}^2 \to \mathbb{R}$ with width 2 depth 2 and bit complexity 2 such that $F\left(\begin{pmatrix} x \\ y \end{pmatrix}\right) = 1$ if $x \in [y, y+1]$ and $F\left(\begin{pmatrix} x \\ y \end{pmatrix}\right) = 0$ if $x > y + \frac{3}{2}$ or $x < y - \frac{1}{2}$.*

*Proof.* We consider the following neural network:

$$F\left(\begin{pmatrix} x \\ y \end{pmatrix}\right) = \sigma(1 - \sigma(2y - 2x)) + \sigma(1 - \sigma(2x - 2y - 2)) - 1 .$$

It is easy to see that this network satisfies the requirements. It has width 2, depth 2 and bit complexity 2. $\qquad \square$

## B  PROOF FROM SECTION 6

*Proof of Theorem 6.2.* We first use Lemma A.2 to construct a network $H : \mathbb{R}^d \to \mathbb{R}$ in the same manner of the construction of $F_1$ is the first stage of Theorem 3.1. This is a 2-layer network with width 1 and bit complexity of $O(\log(R))$. We denote the output of $H$ on the samples $\mathbf{x}_1, \dots, \mathbf{x}_N$ as $x_1, \dots, x_N$. Note that by the construction $|x_i| \leq O(R)$ for every $i \in [N]$ and $|x_i - x_j| \geq 2$ for every $i \neq j$.

We now split the inputs to $\frac{N}{B^2}$ subsets of size $B^2$ each, we denote these subsets as $I_1, \dots, I_{\frac{N}{B^2}}$ (We assume w.l.o.g. that $\frac{N}{B^2}$ is an integer, otherwise we can replace it with $\lceil \frac{N}{B^2} \rceil$). For each subset $I_k$ we use Theorem 3.1 to construct a network $F_k$ which memorizes the points in $I_k$, with the following changes: (1) There is an additional coordinate which memorizes the input and output as is; and (2) The output of the network is added to the output of the network for the previous subset $k - 1$. For $k = 1$, we set this coordinate to be zero. That is, if $x_i \in I_k$ then $F_k\left(\begin{pmatrix} x_i \\ y \end{pmatrix}\right) = \begin{pmatrix} x_i \\ y + y_i \end{pmatrix}$, otherwise $F_k\left(\begin{pmatrix} x_i \\ y \end{pmatrix}\right) = \begin{pmatrix} x_i \\ y \end{pmatrix}$.

Finally, we construct the network $F : \mathbb{R}^d \to \mathbb{R}$ as:

$$F = G \circ F_{\frac{N}{B^2}} \circ \cdots \circ F_1 \circ H ,$$

where $G\left(\begin{pmatrix} x \\ y \end{pmatrix}\right) = y$.

By the construction of each $F_k$ from Theorem 3.1, for every $x \in \mathbb{R}$, if $|x - x_i| \geq 2$ for every $i \in I_k$, then $F_k(x) = 0$. For every $i \in [N]$, there is exactly one $k \in [N/B^2]$ such that $i \in I_k$. Using the

projection $H$, for this $k$ we get that $F_k\left(\begin{pmatrix} x_i \\ y \end{pmatrix}\right) = \begin{pmatrix} x_i \\ y + y_i \end{pmatrix}$, and for every $\ell \neq k$ we get that $F_\ell\left(\begin{pmatrix} x_i \\ y \end{pmatrix}\right) = \begin{pmatrix} x_i \\ y \end{pmatrix}$. This means that $F(x_i) = y_i$ for every $i \in [N]$.

By Lemma A.4 and Lemma A.5, since each $F_k$ is used to memorize $B^2$ samples, then its bit complexity is bounded by $O\left(B\sqrt{\log(B)} \cdot \log(R)\right)$ bits, hence this is also the bit complexity of $F$. The depth of each component $F_k$ is $O\left(B\sqrt{\log(B)} \cdot \log(R)\right)$, and there are $\frac{N}{B^2}$ such components. Hence, the depth of the network $F$ is $O\left(\frac{N\sqrt{\log(B)}}{B}\log(R)\right)$. The width of $F$ is bounded by the width of each component, which is $O(1)$. $\qquad\square$

