# OpenReview forum: "On the Optimal Memorization Power of ReLU Neural Networks"
_ICLR.cc/2022/Conference — ICLR 2022 Spotlight_

### Official Review · Reviewer_FKzU · 2021-10-31

**Correctness:** 4
**Technical Novelty And Significance:** 4
**Empirical Novelty And Significance:** 3
**Recommendation:** 8
**Confidence:** 4

**Main Review:**

I do not have much against the acceptance of this paper. The proof is elegant, the result is significant (providing a matching upper bound, asymptotically), and the writing is clear. It seems likely that the used proof techniques are useful for the proof of relevant theoretical results on the expression power of neural networks. In fact, the results look almost directly applicable to prove similar results on non-ReLU network.

- A suggestion is to include some discussions about the "packing" of separated input data into a bounded input space. In particular, as this paper gives a big-O style bound which is more meaningful when $N \to \infty$ is very large, we should take care of the fact that the separation parameter $\delta > 0$ of the input signals may actually depend on this $N$---one cannot expect to big $\delta$ when $N$ is large. To my understanding, this will not alter the dominant asymptotics of the bound, but providing formal discussion on this matter may help clarify the point.

- One thing I would like to point out is that there are some existing arguments surrounding the total-bit-constraints. In particular, section 4.3 of the COLT-accepted version of Park et al. (2021) gives a similar-yet-informal arguments on the required bit precision of each weight and the necessity of $N$-dependent precision.

- A question that I have is whether one can use a similar technique to utilize the "simplicity of the dataset" to give a better bound. More specifically, note that the $\delta$-separation assumption can be viewed as some way of imposing the assumptions on the underlying data distribution, and it seems to be a widely accepted folklore that the simpler datasets can be expressed with a smaller neural network. As is paper already gives some idea on how to utilize the label information for the construction of partitions, with formal bit-precision arguments, I am curious if authors believe that a similar technique can be used to give a result that shows "under distribution-simplicity assumptions, one requires strictly less than $sqrt(N)$ connections or $N$ bits to memorize the dataset of size $N$."

- In terms of the impact, the result may not give a very powerful message toward practitioners, especially because of the (somewhat depressing) bit-level constraints. However, I expect a long-lasting impact on theoretical society; indeed, this result combined with [Malach et al. (2021)](https://arxiv.org/abs/2102.00434) may indeed lead to the characterization of "memorization threshold," which in turn seems to be an informative quantity in the analysis of generalization capabilities.

- Typo: Page 2, line 15: In his case -> In this case. Page 3, second line of Section 2: Treat is as an integer -> Treat it as an integer. Page 5, first line of "Stage 2", $(x_N,y_n)$ -> $(x_N,y_N)$. Page 12, second line: to proof Theorem 3.1. -> to prove Theorem 3.1.

**Summary Of The Paper:**

The paper shows that, under a mild separability condition, the number of connections in a ReLU network required to memorize N data points is of order $\sqrt{N}$ (up to logarithmic factors). This result improves upon the recent work of Park et al. (2021), and matches the previous lower bound. One key proof-technical innovation is pairing _both the input and label informations_ for each partition and encoding them at stage 2, whereas the previous approach focused on the input only. Authors also refine the framework to incorporate the bit-precision constraints, based on which Telgarsky's bit extraction technique can be used.

**Summary Of The Review:**

The provided results are significant and correct, and the proof technique seems to be useful for other domains of theoretical analyses on neural networks.

---

> ### Author Response · Authors · 2021-11-21
> **Response**
>
> We thank the reviewer for the constructive and positive review, and will make sure to fix the typos. Below we address the main comments.
>
> 1) We will add a more careful discussion on the packing of the data and the dependence of $\delta$ on $N$. In a nutshell, since our bound is logarithmic in $\delta$, then $\delta$ can be polynomial in $N$ and our bound would not change asymptotically.
>
> 2) We will emphasize in the related works section that in Park et al. there is also a discussion about the bit complexity, and the necessity of the dependence on $N$.
>
> 3) Regarding the “simplicity of the dataset”, from the VC dimension bound of Goldberg et al., we know that it is not possible to shatter even a single set of $N$ points (no matter how “simple” they are) with less than $\sqrt{N}$ parameters. In particular, it is not possible to memorize *all* sets of points, even when restricting to simpler datasets. Hence, stronger assumptions on the dataset would not allow improving our upper bound (at least up to log factors).

---

> > ### Comment · Reviewer_FKzU · 2021-11-29
> > **Thank you for the response**
> >
> > Thank you for the response. Although the authors did not provide an updated version (as far as I know), the suggestions I made does not have any critical impact on the core contribution of the manuscript, which is already very nice.

---

### Official Review · Reviewer_P138 · 2021-11-03

**Correctness:** 4
**Technical Novelty And Significance:** 3
**Empirical Novelty And Significance:** Not applicable
**Recommendation:** 8
**Confidence:** 5

**Main Review:**

Strengths:
- The paper proves both an upper and lower bound, improving the best known upper bound (Park et al.) and showing that it is optimal up to log factors.
- The paper is well written and the proof sketches provide sufficient intuition.
- The discussion on related work is sufficient.

Concerns/Comments:
- In the first part of the paper (below Theorem 1.1), there is discussion about why it is surprising that shattering a single set of N points is **not** more difficult than memorizing any set of $N$ points. I don't follow this. The reason VC dimension acts as a trivial lower bound on memorization is that in the former, it suffices to just shatter _any_ set of $N$ points while this does not guarantee memorization of all sets of $N$ points even under some assumptions.
- There appear to be a few minor typos in Section 2. Please take a look.
- I find the definition of bit complexity of the network to be a little misleading. I would prefer if it were stated as the bit complexity of the weights of the network since one would intuitively expect the bit complexity of the network to be the bit complexity of each weight times the number of weights.
- In section 4, I think $r=1$ or $\delta^{-1} \geq 2(N-1)$. It's just a constant off, so this is a minor comment.
- In the appendix, it is mentioned that ceil's can be applied in many instances where the $\log(\cdot)$ terms may not be integral. Again, a minor comment but it would be useful to mention this in the preliminaries instead.
- The proofs of the two lower bounds (for general $L$ and $L=\sqrt{N}$) use different VC dimension bounds (Golbderg et al. and Bartlett et al.). Can you clarify the relationship between this two bounds? Is one tighter than the other in general? It seems to me that the Bartlett bound is tighter if there are no restrictions on $L$. In the discussion below Theorem 5.1, if I plug in $L=N$, then the VC dimension gives us a trivial lower bound of $W = \tilde{\Omega}(1)$ which doesn't seem to make sense. Can you clarify?

**Summary Of The Paper:**

The paper shows that ReLU networks can memorize $N$ points using $\tilde{O}(N)$ parameters as long as they are $\delta-$separated.  They also prove a lower bound on number of parameters and bit complexity showing that this is optimal.

**Summary Of The Review:**

Overall, I found the paper to be very well written and an excellent contribution! I recommend that it be accepted.

---

> ### Author Response · Authors · 2021-11-21
> **Response**
>
> Thank you for the useful comments. We are glad that you appreciate the contribution of the paper.
> We will fix the typos and minor comments. Below we address the main questions that were raised.
>
> 1) Regarding the first comment: Right, we meant to write it the other way around, and will fix it in the camera ready version.
>
> 2) We will modify the definition of bit complexity, to be the bit complexity of the weights, and the bit complexity of the network to be the bit complexity of the weights times the number of weights
>
> 3) The difference between the two VC dimension lower bounds: Goldberg et al. give a VC bound independent of the depth which implies that shattering $N$ points requires $\Omega(\sqrt{N})$ parameters. The Bartlett et al. bound implies that shattering $N$ points requires $\Omega\left(\frac{N}{L \log(N)}\right)$ parameters. The latter bound is tighter when the depth is bounded. For example, for networks of depth $O(1)$, the bound from Bartlett et al. is $\tilde{\Omega}(N)$ parameters for shattering $N$ points, which is tighter than Goldberg et al. On the other hand, for large values of $L$ the bound of Bartlett et al. is less tight. Note that in Theorem 5.1 we only consider $1 \leq L \leq \sqrt{N}$.

---

> > ### Comment · Reviewer_P138 · 2021-11-27
> > **Response to rebuttal**
> >
> > Thank you for clarifying, that makes perfect sense.

---

### Official Review · Reviewer_sa2e · 2021-11-05

**Correctness:** 3
**Technical Novelty And Significance:** 4
**Empirical Novelty And Significance:** Not applicable
**Recommendation:** 8
**Confidence:** 5

**Main Review:**

1. Overall, I strongly recommend accepting this paper. It improves upon the existing result by Park et al. (2020), where memorization of $N$ points by neural networks with $\tilde \Theta(N^{2 / 3})$ parameters was shown. The paper improves the upper bound on the number of parameters required to memorize $N$ points from $\tilde O(N^{2 / 3})$ to $\tilde O(N^{1 / 2})$ and closes the gap between the lower bound $\Omega(N^{1 / 2})$.

2. Another noteworthy strength of this submission is that the paper gives a careful treatment of the bit complexity. Many existing memorization results, even those that require $N$ parameters, assume infinite precision of network parameters. While these results provide valuable insights, they always face criticisms such as "look, if you assume infinite precision, then you can memorize any dataset in a single real number, because there exists a bijection from $\mathbb R$ to $\mathbb R^2$!" By carefully bounding the number of bits required, this paper constructs a memorizing ReLU network that is not only optimal in the number of parameters but also optimal in the number of bits used in the weights.

3. When counting the number of parameters, the paper does NOT count the zero weights in the network as parameters, rather than "(# of parameters) = (# of entries in weight matrices and bias vectors)." This convention of allowing *disconnected edges* in fully-connected networks was also used in the VC-dimension lower bound construction (which involves a similar bit extraction process) in Bartlett et al. (2019). The paper exploits this convention in Theorem 5.1, where they construct $\frac{N}{L^2}$ parallel subnetworks where each of the subnetworks is constructed using Theorem 3.1. Even though this way of construction results in a width-$O(\frac{N}{L^2})$ network, the width does not introduce $O(\frac{N^2}{L^4})$ parameters between each pair of hidden layers, because zero weights are not counted as parameters. While I believe ignoring zero weights is a reasonable thing to do, this allows a lot of additional freedom when constructing memorizers. It is important to note that some existing results (e.g. Park et al. (2020)) do not rely on this counting convention in their constructions.

4. I am a little confused on the bit complexity results presented in Section 6, maybe because I'm not an expert on bit complexity theory. In Theorem 6.1, it is assumed that "each parameter is represented by $B$ bits," but in the proof it is stated that "each weight of f can be represented by at most $B+2\log(U)$ bits." What is the difference between *representing a parameter* versus *representing a weight*? What do we mean exactly by *representing* something with a certain number of bits? Should the representation of a matrix entry include its index or not? A little more formal definition of these notions should prove helpful.

5. I think the paper will benefit from a little more detailed discussion in the proof intuition section. Allow me to elaborate:

5a. Park et al (2020) also use some bit extraction techniques inspired from Bartlett et al (2019). What is the key insight/trick/technique in this paper that allows a reduction from $N^{2/3}$ to $N^{1/2}$?

5b. More details on depth and bit complexity of parameters would be beneficial. The sketch discusses the bit complexity of $w_j$ and $u_j$'s, but does not say that (unless I missed it) these crafted integers are used as *parameters* in the constructed network; hence, the connection between the bit complexities of these integers and the network parameters are left somewhat unclear.

6. In the discussion of lower bounds, I question if Sontag (1997) really gives a lower bound of $\Omega(N)$ for *deep* ReLU networks? If I understood correctly, the results in Sontag (1997) on "piecewise analytic definable" function class (Corollary 4.1) requires that the partition of the parameter/input space (which is introduced by ReLU hidden nodes) must be determined by analytic definable functions (see Page 6 of Sontag (1997)). For 1-hidden-layer ReLU networks the result can be applied, but I don't think this result applies to ReLU networks of more than 1 hidden layer? In any case, in my view, the key to achieving memorization with $o(N)$ parameter seems to lie on $L$ increasing with $N$, rather than the separation assumption on data points.

7. Can the authors comment on extension to other activations? It seems that the main reason why the paper limits itself to ReLU activation is due to the Telgarsky triangle used in the bit extraction process (Lemma A.7). While I like the elegant use of the Telgarsky triangle in the lemma, I reckon there may be some other ways to extract bits sequentially without having to rely on ReLU and Telgarsky triangle.

8. Minor suggestions:
- Maybe change Lemma 4.1 to Proposition 4.1, since it is not used as a subroutine for other theorems? Also, maybe consider replacing the exponent $\epsilon$ with some other character because it is already used in Remark 3.3 as regression accuracy.
- Proof of Lemma A.4: In the beginning, $F_j : \mathbb R^2 \to \mathbb R^2$
- Proof of Lemma A.5: Typo in $F_i$, $2^{n\cdot \rho}+2$ -> $2^{n\cdot \rho+2}$. Also in the sentence "We use Lemma A.8 with inputs $BIN_{i\cdot\rho+1:(i+1)\cdot\rho}(u)$ and $x$", consider switching $BIN_{\dots}$ and $x$ in the sentence because $BIN_{\dots}$ is the one that is substituted to $y$.
- Lemma A.7: This is very minor, but why necessarily $i < j$? It looks to me that $i = j$ can be allowed?
- Proof of Lemma A.7: $\psi^{(i)}$ has $2^{i-1}$ bumps of slope $\pm 2^i$? The triangle $\psi$ has one bump.


**Summary Of The Paper:**

This paper studies memorization capacity of deep ReLU networks. For arbitrary $N$ data points in a ball of size $r$ satisfying minimum separation $\delta$, the authors show that there exists a ReLU network of constant width and depth $\tilde O(\sqrt{N})$ that perfectly memorizes the entire dataset (Theorem 3.1). This means that memorizing arbitrary $N$ data points can be done using only $\tilde O(\sqrt{N})$ parameters, when depth increases with $N$. Combined with a classical upper bound on VC dimension (Goldberg and Jerrum (1995)), the construction is optimal up to log factors. Theorem 3.1 is extended to the case of fixed depth $L \leq \sqrt{N}$ (Theorem 5.1) and fixed bit complexity per parameter (Theorem 6.2), and these additional results are also optimal modulo log factors.

**Summary Of The Review:**

I strongly recommend accepting this paper. The paper provides a tight construction that shows the optimal number of parameters in a ReLU network required to memorize arbitrary $N$ data points (comment 1). It also carefully discusses bit complexity and shows that the construction is also optimal in terms of the number of bits required (comment 2). For the rest of the comments (3-8), I raise some minor criticisms/questions/suggestions, but I believe the authors will clarify my concerns.

---

> ### Author Response · Authors · 2021-11-21
> **Response**
>
> We thank the reviewer for the thorough and positive review, and we will fix the typos and minor comments accordingly. Below we address the main questions that were raised.
>
> 4. Regarding the bit complexity result in section 6, in the proof of Theorem 6.1 we meant to view f as a directed graph, and each edge can be represented by $B+2\log(U)$ bits ($2\log(U)$ bits to represent the indices of the neurons that the edge connect, and $B$ bits for the weight itself). We agree that the phrasing is a bit confusing and we will clarify it in the camera ready version.
>
> 5a. Regarding the intuition for the proof of the main result, in the last paragraph of Section 3 we discuss the main conceptual difference between our methods and prior results. Namely, in stages II and III in our proof, we also encode the inputs as weights of the network (in the $u_j$ parameters). We will emphasize that in the camera ready version.
>
> 5b. We will clarify this issue.
>
> 6. We agree that the key to achieve memorization with $o(N)$ parameters, both in our result and in Park et al., is that $L$ grows with $N$. Nevertheless, both in our result and in Park et al. the separability assumption was necessary. We will rephrase the reference to Sontag (1997) to be more accurate.
>
> 7. We believe that our results can be extended to other piecewise linear activations which allow efficient bit extraction, either with Telgarsky’s triangle function or with the techniques from Bartlett et al. 2019. Such activations can include for example STEP + ID and hard tanh.

---

> > ### Comment · Reviewer_sa2e · 2021-11-28
> > **Response acknowledged**
> >
> > I thank the authors for their response, which addressed most of my comments and questions.
> >
> > It would have been even better if they provided a revised manuscript, but they did not, as far as I can tell. I believe that the authors will incorporate the reviewers' comments accordingly in the camera-ready version.
> >
> > In the revision, I also do hope that the authors add an appropriate comment on the point 3 that I raised; I believe that it is important to note the distinction between results that "ignore zero-weight connections" and the ones that don't.
> >
> > Best,
> >
> > Reviewer sa2e

---

> > > ### Author Response · Authors · 2021-11-30
> > > **Re: Response**
> > >
> > > We thank the reviewer for the comment regarding the zero weight connections, and we agree this is an important distinction to be noted. We will emphasize this in the camera-ready version. We also want to note that in our main result (Theorem 3.1) and the results in Section 6, we count the number of parameters as the number of weights in the network, i.e. not ignoring zero weights. As the reviewer correctly stated, in Theorem 5.1 we do ignore zero weights to get a tight comparison to the VC dimension bound from Bartlett et al., where they also ignored zero weights for their result.

---

### Decision · Program_Chairs · 2022-01-20

**Decision:**

Accept (Spotlight)

**Comment:**

This paper studies the memorization power of Relu Neural networks and obtains sharp bounds in terms of parameters. The writing is very clear and the results very interesting.